# Bayesian Inverse Transition Learning for Offline Settings

Leo Benac [1]   Sonali Parbhoo [2]   Finale Doshi-Velez [1]

## Abstract

Offline Reinforcement learning is commonly used for sequential decision-making in domains such as healthcare and education, where the rewards are known and the transition dynamics $T$ must be estimated on the basis of batch data. A key challenge for all tasks is how to learn a reliable estimate of the transition dynamics $T$ that produce near-optimal policies that are safe enough so that they never take actions that are far away from the best action with respect to their value functions and informative enough so that they communicate the uncertainties they have. Using an expert's feedback, we propose a new constraint-based approach that captures our desiderata for reliably learning a posterior distribution of the transition dynamics $T$ that is free from gradients. Our results demonstrate that by using our constraints, we learn a high-performing policy, while considerably reducing the policy's variance over different datasets. We also explain how combining uncertainty estimation with these constraints can help us infer a partial ranking of actions that produce higher returns, and helps us infer safer and more informative policies for planning.

## 1. Introduction

In standard planning scenarios, one is given the rewards $R$ and the transition dynamics of the environment $T$ and asked to compute the optimal set of actions or policy $\pi$. However, in many real settings, the transition dynamics $T$ is not available. In such settings, model-based (Reinforcement Learning) RL—that is, first learning the transition dynamics $T$ and then planning—is useful because it can help us be more data efficient, simulate data, encourage exploring,

capturing important details of the environment as well as counterfactual reasoning (Ha & Schmidhuber, 2018; Oh et al., 2015; Buesing et al., 2018).

In this paper, we are particularly interested in learning the transition dynamics $T$ in offline settings, that is from already-collected batch data. Such settings are common in healthcare and education. However, learning the dynamics $T$ is a hard problem as the number of parameters to estimate grows with the dimensions of state and action spaces. There is a tension between a model class for the dynamics that is expressive enough to capture what is needed and one that is small enough to learn (Zhu et al., 2023; Ayoub et al., 2020). Moreover, in offline settings, we cannot interact with the environment to generate more data; we are limited to the exploration (or lack thereof) to that the users who produced the data (Herman et al., 2016).

Our goal in this work is to take advantage of the fact that, in many scenarios, the users who generated the trajectories of batch data can be presumed to be near-optimal—or at the very least, not highly sub-optimal. We consider a version of the learning using feedback from an expert through demonstrations that we call *Inverse Transition Learning (ITL)*. Here, we presume that both the rewards $R$ and a near-optimal policy $\pi_b$ are available to us; our goal is to estimate a posterior distribution of the dynamics $T$ (either for further optimization of the policy or for use in planning against other rewards $R'$). In later versions, we will relax the requirement that the user policy $\pi_b$ is given.

Broadly, our approach consists of three parts: First, we use feedback from the expert to create a set of constraints that require that the value of actions taken by the expert user is higher than the values of actions they never took. Next, we create a gradient-free way to estimate the posterior distribution over the dynamics $T$ in a way that satisfies those constraints. This process guarantees that policies associated with planning against *any* sample of the dynamics $T$ will be highly performant—potentially better than the original expert—even with limited coverage of the state-action space. We then demonstrate how planning against the maximum likelihood estimate of the dynamics $T^{MLE}$ often results in policies that have high variance depending on the particular sample of trajectories in the offline batch. Then, we show that having a posterior distribution over the dynamics $T$ and

[1]Department of Computer Science, Harvard University, Cambridge, MA, USA [2]Department of Engineering, Imperial College London, London, UK. Correspondence to: Leo Benac <lbenac@g.harvard.edu>, Sonali Parbhoo <s.parbhoo@imperial.ac.uk>, Finale Doshi-Velez <finale@seas.harvard.edu>.

*Interactive Learning with Implicit Human Feedback Workshop at ICML 2023.*

also a posterior over potential optimal policies results in higher performing and more informative policies, as well as lower variance across different batches of data.

## 2. Related Work

**Learning from Demonstrations** Learning behavior policies from demonstration trajectories produced by a near-optimal expert through supervised learning is called *Imitation Learning* (IL). Different IL methods have been developed. (Ross et al., 2010) trains a policy at each time step, (Ross et al., 2011) introduces an iterative algorithm DAgger where at each iteration the agent needs to ask an optimal expert a set of actions for some states that the agent explored, then aggregate the dataset with this new information and retrain a policy on the aggregated dataset. (Kim et al., 2013) introduces Approximate Policy Iteration with Demonstration (APID), where they combine expert data and RL signals and show that the policy that they learn is robust to suboptimal demonstrations.

Outside of learning the policy, learning the reward function $R$ may be of interest as it gives a succinct description of a task, and such reward function $R$ can be used for planning and transfer learning. (Ng et al., 2000) introduce *Inverse Reinforcement Learning* (IRL) where the focus is on learning the reward function $R$ given T or access to a simulator to do roll-outs, as well as near-optimal demonstrations from an expert and in some cases its policy. (Abbeel & Ng, 2004) show that learning the reward function $R$ through the principle of maximum margin can be used to learn the behavior policy. (Ziebart et al., 2008) show how the principle of Maximum Entropy can be used to learn the reward function $R$ and corresponding policies that are robust to suboptimal demonstrations. (Ramachandran & Amir, 2007) solves the non-identifiability of the reward function $R$ by learning a posterior distribution rather than a point estimate.

ITL focuses on learning the transition dynamics $T$ given $R$, expert demonstrations, and its policy. Using Bayesian inference, we propose to learn a posterior distribution of the transition dynamics $T$ in an offline setting. Every sample of this distribution is guaranteed to produce a near-optimal policy downstream. The most closely related among these works is (Herman et al., 2016) where the authors present a gradient-based IRL approach for simultaneously estimating system dynamics and rewards via combined optimization in tabular MDP's. However, (Herman et al., 2016) does not explicitly explore the relation between the estimated and true dynamics and learn the dynamics $T$ by gradient-based optimization. In contrast to these methods, we i) present a constraint-based optimization that is free of gradients to guarantee learning a posterior distribution of $T$ that yields a performant and informative policy. We not only learn the system dynamics but also characterize where methods

such as $T^{MLE}$ fail under poor sample coverage as this is a common way to estimate the transition dynamics $T$ in such offline settings (Zhang et al., 2021). Other works (Reddy et al., 2018; Golub et al., 2013) developed gradient-based methods to learn the expert's belief of the dynamics, where such beliefs are suboptimal, whereas, in our setting, the expert is uncertain what the best action is.

**Constraint-based RL** Several algorithms study IRL by imposing constraints on the rewards to induce, for instance, safety. For instance, Fischer et al. (2021) presents Constrained Soft Reinforcement Learning (CSRL), an extension of soft reinforcement learning to Constrained Markov Decision Processes (CMDPs) where the goal of the reward maximization is regularized with an entropy objective to ensure safety. (Scobee & Sastry, 2019) introduce an approach based on the Maximum Entropy IRL framework that allows inferring which constraints can be added to the MDP to most increase the likelihood of observing these demonstrations. Our work may be seen as a specific form of constraint-based RL: we utilize observational data from an expert agent to create a set of constraints and perform inference on $T$ within a discrete MDP setting. In contrast to these approaches, we leverage (near) optimal data to estimate transitions for actions that lack data, ultimately resulting in a more accurate $T$ estimate than what traditional maximum likelihood estimates would produce.

## 3. Preliminaries

**Markov Decision Processes (MDPs)** An MDP $M$ can be respresented as a tuple $M = \{S, A, T, \gamma, R\}$, where S is the state space, $A$ is the action space, $T$ is the dynamics of the environment, $\gamma$ is the discount factor and $R$ is a bounded reward function. In RL the goal is to find the best policy $\pi^*$ corresponding to a MDP M. In a discrete MDP there exists at least one optimal deterministic policy. We measure the quality of a policy $\pi$ by looking at its corresponding value functions. The value functions $V^\pi$ and $Q^\pi$ are the expected cumulative reward by taking actions with respect to a policy $\pi$. $V^\pi(s)$ and $Q^\pi(s, a)$ will both start from state s but $Q^\pi(s, a)$ will take action a before following $\pi$. They can be computed through the Bellman Equations below, where $r_t$ is the reward at time $t$:

**Bellman Equations** The value of a policy $\pi$ at state $s$ is given by the expected sum of rewards,

$$
\begin{aligned}
V^\pi(s) &= \mathbb{E}[\sum_{t=0}^{\infty} \gamma^t r_t | s_0 = s, \pi] \qquad (1) \\
&= \mathbb{E}_a[R(s, \pi(a|s))] + \gamma \mathbb{E}_{s'}[V^\pi(s')]
\end{aligned}
$$

The value of a policy $\pi$ at state $s$ when performing action $a$

is given by,

$$Q^\pi(s,a) = \mathbb{E}[\sum_{t=0}^{\infty} \gamma^t r_t | s_0 = s, a_0 = a, \pi] \quad (2)$$

$$= R(s,a) + \gamma \mathbb{E}_{s'}[V^\pi(s')] \quad (3)$$

**Notation:** Let $V^\pi$ denote the vector of values $V^\pi(s)$. We use shorthand $R_a, Q_a,$ and $T_a$ to represent the vectors $R(\cdot, a)$ and $Q(\cdot, a)$ and the matrix $T(\cdot|\cdot, a)$. We also use use shorthand $R_\pi, Q_\pi,$ and $T_\pi$ to represent the vectors $\mathbb{E}_{a \sim \pi}[R_a]$, $\mathbb{E}_{a \sim \pi}[Q_a]$ and the matrix $\mathbb{E}_{a \sim \pi}[T_a]$. $Q^*$ represents the optimal Q functions in the True unknown environment T (ie. assuming the agent always takes the best action possible $a^*$, where $a^*$ may change depending on the state s that we are in).

In the tabular setting the value functions can be calculated directly through a closed-form solution of $V^\pi$,

$$V^\pi = R_\pi + \gamma T_\pi V^\pi = (I - \gamma T_\pi)^{-1} R_\pi, \quad (4)$$

and $Q^\pi$ respectively,

$$Q_a^\pi = R_a + \gamma T_a V^\pi = R_a + \gamma T_a (I - \gamma T_\pi)^{-1} R_\pi. \quad (5)$$

## 4. Methodology

In this section, we describe the setup of our methods, introduce two key definitions relating the sub-optimality of the expert to $T$, and develop a set of constraints for both an optimal and sub-optimal expert. These constraints enable us to impose the structure we would like to have when estimating $T$. Based on these constraints, we demonstrate how one can infer a reasonable estimate posterior on $T$.

**Problem Setup** We assume we know everything about the MDP $M$ except the true transition dynamics $T$ (i.e. $M \setminus \{T\}$). We assume we are also given an expert policy $\pi_b$ and offline tabular observational batch data $\mathcal{D}$, where $\mathcal{D}$ is assumed to have been generated by rollouts of $\pi_b$ with respect to the true unknown transition dynamics $T$. We assume that the policy of the expert we are given, $\pi_b$ is $\epsilon$-optimal.

**Definition 4.1.** (*$\epsilon$-ball*) For a state s, an action a belongs to the $\epsilon$-ball $\epsilon(s; T')$ if the action-value of that action $Q(s,a)$ is within $\epsilon$ of the optimal action-value $\max_{a'} Q(s, a')$ with respect to the transition dynamics $T'$.

**Definition 4.2.** (*$\epsilon$-ball property*) We say that a point estimate $\widehat{T}$ has the $\epsilon$-ball property with respect to the true transition dynamics $T$ if they have the same $\epsilon$-balls for every state s : $\epsilon(s; \widehat{T}) = \epsilon(s; T)$.

Having a transition dynamics estimate $\widehat{T}$ that has the $\epsilon$ ball property is important as we want to make sure that it captures the same structure as the true transition dynamics $T$

(ie. for the states where the expert user is certain of the best action $a^*$ have the value of $a^*$ at least $\epsilon$ away from the other actions and for the uncertain states have the actions selected by the user expert $\epsilon$ close to each other, all with respect to $\widehat{T}$).

**Definition 4.3.** (*$\epsilon$-optimality*) A policy $\pi$ is $\epsilon$-*optimal* if it only takes actions in the $\epsilon$-ball for all states s with respect to the true transition dynamics $T$, that is, only takes actions in $\epsilon(s; T)$.

We assume that the expert policy $\pi_b$, which is $\epsilon$-optimal, is uniform in each of the actions in the $\epsilon$ ball $\epsilon(s; T)$, where $T$ is the true unknown dynamics. This definition of sub-optimality is similar to how clinicians would behave in real healthcare settings, for states where there is a clear best treatment it makes sense for them to only pick that one treatment but for other states where more than one treatment seems appropriate it makes sense to assume complete uncertainty among these treatments if no else prior information is given.

**Definition 4.4.** (*deterministic/stochastic-policy state* ) A deterministic-policy state is a state for which the expert $\pi_b$ takes only one action (ie. a state for which we know the best action to take since we assume we are given the policy of the expert $\pi_b$) and a stochastic-policy state is a state for which the expert takes multiple actions (i.e. a state for which the expert has uncertainty over a certain group of actions).

Note that when $\epsilon = 0$, the expert is fully optimal and hence we would only get deterministic-policy states, whereas as $\epsilon$ gets bigger we start having more stochastic-policy states. A common method to estimate the dynamics $T$ in offline settings is to use MLE estimates. However since our batch data $\mathcal{D}$ only contains expert demonstrations, there can be states and action pairs $(s, a)$ for which there is no data. A common way around that is to assume we see every transition at least once before computing the MLE estimates, which is also called smoothing (Zhang & Teng, 2021) . We refer to this as $T^{MLE}$. It is the equivalent of assuming a uniform prior on $T$ and setting $T^{MLE}$ to be the mean of this posterior, assuming a Dirichlet-Multinomial Probabilistic Model over $T$ (ie. $P(T|Data)$), which we also refers to it as the Unclipped Posterior:

**Probabilistic Model over T (Unclipped Posterior)**

$$
\begin{aligned}
\text{Prior}: \quad & Dir(\mathbf{1}|s,a) \\
\text{Likelihood}: \quad & Multinomial(N_{s,a}|s,a) \\
\text{Posterior}: \quad & Dir(N_{s,a} + \mathbf{1}|s,a)
\end{aligned}
$$

$N_{s,a}$ and $\mathbf{1}$ are both vectors of dimensions $|S|$. $N_{s,a}$ is the number of transitions in the batch data from state $s$ and action $a$. Using the assumptions we have made, we can develop a set of constraints based on the expert policy $\pi_b$

that highlights what properties we want our $T$ to satisfy by using the Closed Form Bellman Equations for tabular MDPs.

## 4.1. Constraints on $T$ given that the expert policy $\pi_b$ is $\epsilon$-optimal

We construct two different sets of constraints that $T$ should satisfy. The first set of constraints is applicable when the expert is fully optimal, and the second when the expert is suboptimal. $\widehat{T}_{\pi_b}$ in the constraints below could be sampled from the posterior of the data model or we could use $\widehat{T}_{\pi_b} = T_{\pi_b}^{MLE}$. These are both reasonable choices since this is part of the state action space for which we have adequate data.

**Fully Optimal Expert Policy Constraints** When $\pi_b = \pi^*$,

$$T_{a^*}(I - \gamma\widehat{T}_{\pi_b})^{-1}R_{\pi_b} = \frac{1}{\gamma}((I - \gamma\widehat{T}_{\pi_b})^{-1}R_{\pi_b} - R_{a^*}) \quad (6)$$

and for $a \neq a^*$

$$T_a(I - \gamma\widehat{T}_{\pi_b})^{-1}R_{\pi_b} < \frac{1}{\gamma}((I - \gamma\widehat{T}_{\pi_b})^{-1}R_{\pi_b} - R_a - \epsilon) \quad (7)$$

In what follows, we show that if $T$ satisfies the constraints above, then $T$ will capture the correct structure provided by the $\epsilon$-optimal expert.

**Lemma 4.5.** *If $T$ satisfies the above constraints then $T$ will recover the $\epsilon$-ball property. That is, let $a$ be any action such that $a \neq a^*$. Then,*

$$T_a(I - \gamma\widehat{T}_{\pi_b})^{-1}R_{\pi_b} < \frac{1}{\gamma}((I - \gamma\widehat{T}_{\pi_b})^{-1}R_{\pi_b} - R_a - \epsilon)$$

$$\Longleftrightarrow$$

$$R_a + T_a\gamma V^{\pi_b} + \epsilon = Q_a^{\pi_b} + \epsilon < V^{\pi_b} = Q_{a^*}^{\pi_b}$$

$$(8)$$

Proof sketch: We used the fact that if $T$ satisfy the constraints then $T_{\pi_b} = \widehat{T}_{\pi_b}$ (since $\pi_b = a^*$). Hence we see that having a $\widehat{T}$ that satisfies these constraints will result in the optimal action being at least $\epsilon$ better with respect to the value functions with respect to that $\widehat{T}$ and hence recover the $\epsilon$-ball property. □

Lemma 4.5 implies that if $T$ satisfies the constraints then it will always recover the best actions in deterministic-policy states as well as never inducing actions outside of the $\epsilon$-ball in stochastic-policy states. This is why we observe 100% accuracy of our method in the Deterministic and $a \in \epsilon$-balls columns in Table 1.

**Suboptimal Expert Policy Constraints:** For $(s, a^*)$ i.e. where the expert is deterministic:

$$T_{a^*}(I - \gamma\widehat{T}_{\pi_b})^{-1}R_{\pi_b} = \frac{1}{\gamma}((I - \gamma\widehat{T}_{\pi_b})^{-1}R_{\pi_b} - R_{a^*}) \quad (9)$$

For $(s, a)$ such that a $\notin \epsilon(s)$ i.e. actions that the expert never takes:

$$T_a(I - \gamma\widehat{T}_{\pi_b})^{-1}R_{\pi_b} < \frac{1}{\gamma}((I - \gamma\widehat{T}_{\pi_b})^{-1}R_{\pi_b} - R_a - \delta_{s,a})$$

$$(10)$$

Finally, for $(s, a)$ where $a \in \epsilon(s)$:

$$T_a(I - \gamma\widehat{T}_{\pi_b})^{-1}R_{\pi_b} < \frac{1}{\gamma}((I - \gamma\widehat{T}_{\pi_b})^{-1}R_{\pi_b} - R_a + \delta_{s,a})$$

$$(11)$$

and

$$\frac{1}{\gamma}((I - \gamma\widehat{T}_{\pi_b})^{-1}R_{\pi_b} - R_a - \delta_{s,a}) < T_a(I - \gamma\widehat{T}_{\pi_b})^{-1}R_{\pi_b}$$

$$(12)$$

If $\pi_b$ is not fully optimal, meaning there are states for which the expert takes more than 1 action (stochastic-policy states), we can still develop similar constraints and guarantees. However in this case, if $T$ satisfies the Suboptimal Expert Policy Constraints above, $T$ will not necessarily have the $\epsilon$-ball property due to the nonlinearity of the actions in the stochastic-policy states. We can however still make sure that the $T$'s satisfying the constraints have the $\epsilon$-ball property by tuning the $\delta_{s,a}$ appropriately, where $\delta_{s,a}$ are constants. Empirically, when $\delta_{s,a} = \epsilon$ does not seem to output $T$'s that have the $\epsilon$-ball property, it seems to be a good heuristic to make the constraints tighter by increasing/decreasing $\delta_{s,a}$ in the inequality 10/ 11&12 respectively.

## 4.2. Estimating the Posterior over $T$ given that the expert is near-optimal

Using the batch data $\mathcal{D}$, we can construct a posterior distribution over $T$, $P(T|Data)$, that we will refer to as *Unclipped*. $T^{MLE}$ is the mean of this posterior. However, the problem with such estimates of $T$'s is that they do not take advantage of the fact that the expert is acting near-optimally. The two sets of constraints we developed impose $T's$ satisfying such constraints to have the $\epsilon$-ball property. We use those constraints to "Clip" the posterior distribution through rejection sampling and model $P(T|Data, \text{expert is } \epsilon\text{-optimal})$. The idea is to sample from the posterior distribution constructed from the data (Unclipped) but to only accept the samples that satisfy our constraints. This leaves us with an empirical distribution of $T$'s that recover the $\epsilon$-ball property). We present an algorithm to clip the Unclipped posterior in Algorithm 1. We call this resulting posterior distribution *Clipped*.

**Algorithm 1** CLIP Posterior Rejection Sampling

---

**Inputs**: R, $\pi_b$ ($\epsilon$ expert policy), Posterior $T(.|s,a)$ for each (s,a), Number of samples N required

**Output**: Empirical CLIPPED Posterior of $T(.|s,a)$ for each (s,a)

**While** *samples accepted* $< N$

    Sample $\widehat{T}_{s,a} \sim Posterior(s'|s,a)$ for each (s,a)

    $constraint_a = V_{\pi_b} - (R_a + \widehat{T}_a \gamma V_{\pi_b})$

    **for each (s,a) where** $\pi_b(a|s) = 0$

        **While** $constraint_a(s) < \epsilon$

            $\widehat{T}_{s,a} \sim Posterior(s'|s,a)$

            $constraint_a = V_{\pi_b} - (R_a + \widehat{T}_a \gamma V_{\pi_b})$

    **for each (s,a) where** $\pi_b(a|s) = 1$

        **While** $constraint_a(s) \neq 0$

            $\widehat{T}_{s,a} \sim Posterior(s'|s,a)$

            $constraint_a = V_{\pi_b} - (R_a + \widehat{T}_a \gamma V_{\pi_b})$

    **for each (s,a) where s is a stochastic-policy state and** $\pi_b(a|s) > 0$

        **While** $constraint_a[s] < -\delta_{s,a}$

        $\widehat{T}_{s,a} \sim Posterior(s'|s,a)$

            $constraint_a = V_{\pi_b} - (R_a + \widehat{T}_a \gamma V_{\pi_b})$

        **While** $constraint_a[s] > \delta_{s,a}$

        $\widehat{T}_{s,a} \sim Posterior(s'|s,a)$

            $constraint_a = V_{\pi_b} - (R_a + \widehat{T}_a \gamma V_{\pi_b})$

    **If** $\widehat{T}$ **recover the** $\epsilon$ **ball property**

        Accept $\widehat{T}$

    **Else**

        Tune the different $\delta_{s,a}$ and start over.

Construct CLIPPED Posterior from all of the accepted samples

---

## 5. Experimental Setup

In our experiments, we look at how the $T^{MLE}$ performs across different data and optimality settings, characterize its mistakes as well as highlight certain issues that can arise when learning under expert data. We show how combining our constraints with uncertainty helps our method avoid and reduce such mistakes and pathologies and hence performs better across different metrics and settings. We also show that our constraints considerably reduce the variance of the policy we get over different batch data. Finally, we demonstrate why combining uncertainty with constraints can help us infer a ranking of actions in the $\epsilon$-balls over the stochastic-policy states which results in having more informative, and higher-performing policies when doing planning.

**Setup** For all the results we will show, we used a fixed MDP of 15 states (+1 terminal state) and 6 actions so that it is computationally feasible to run various experiments and big enough so that it is an environment interesting enough. The True $T$ is created such that transitioning from one state to the next state is sometimes uniform and sometimes very skewed towards one or a couple of states so that we have an environment that has a variety of behaviors. We use $\gamma = 0.95$ and a reward function that is action dependent and such that the range of possible values depends on the number of states, $|S| = 15$.

**Generating batch data** $\mathcal{D}$. For $\mathcal{D}$, we simulate episodes as follows: first, we pick a random state among the 15 possible states; next, we roll out the state using $\pi_b$ in the true environment $T$ until we reach the terminal state or until we reach 20 steps [1] This procedure is repeated for $K$ episodes. Based on $\mathcal{D}$, we examine two different settings namely, a low data setting ($K = 15$) and a high data setting ($K = 300$), as well as 3 different values of $\epsilon$ to get an idea of how the task of estimating $T$ varies across a different kind of dataset as well as different degree of optimality: $\epsilon = 0$ (corresponding to a fully optimal expert with 0 stochastic-policy states); $\epsilon = 3$ (3 stochastic-policy states); $\epsilon = 4$ (6 stochastic-policy states)

**Baselines and Metrics.** First, We compare the performance of our approach (Clipped Posterior) to the $T^{MLE}$ and the original (Unclipped) Posterior distribution. We look at the accuracy of deterministic-policy states, and the accuracy of stochastic-policy states, as well as how often. Next, we look at how different methods perform at recovering the best action over states, through planning on their respective estimate of $T$ by computing $Q^*_{metric}$.

$$Q^*_{metric}(\widehat{\pi}) = \sum_{s \in S} \left( Q^*(s, a^*) - \sum_{a \in A} \widehat{\pi}(a|s) Q^*(s, a) \right)$$

This metric gives us a more quantitative idea of how good, the different methods are at estimating a $T$ that captures the best action when planning on such $T$, as well as quantifying the importance of each mistake with respect to the True environment. The better an action is, the lower $Q^*_{metric}$ will be, and the worse an action is the higher $Q^*_{metric}$. We want $Q^*_{metric}$ to be as close as possible to 0 (using $\pi^*$ will achieve such a result). In real life we would not be able to compute such metrics, this is purely for analysis purposes. We also report the result of the $Q^*_{metric}$ for the policy of the expert $\pi_b$. (Note that this policy is not obtained through planning on any $T$ estimate, we just report it to get an idea of how it performs compared to others. It is expected that this $\pi_b$ performs very well since it is $\epsilon$ optimal by definition).

**Training Details** All of the results presented are averaged over 1000 datasets. For the MLE results, we do planning

---

[1]Trajectories of longer than 20 steps are truncated to 20 steps to reflect most realistic healthcare settings.

on $T^{MLE}$ to get $\widehat{\pi}$ and compute the results whereas for the clipped and unclipped results, we do planning on 1000 different $T$'s (sampled using the corresponding distribution) and get 1000 corresponding $\widehat{\pi}$'s, then take the mean. For each Batch Data $\mathcal{D} \in \{\mathcal{D}^{(i)}\}_{i=1}^{1000}$ we do the following:

**MLE Method**
$$\widehat{T} = T^{MLE} \underset{\text{Value Iteration}}{\longrightarrow} (\pi^{MLE}, Q^{MLE}) = (\widehat{\pi}, \widehat{Q})$$
$$\underset{\text{Compute Results}}{\longrightarrow}$$

**Clipped or Unclipped Posterior Method**
$$\{\widehat{T}^{(i)}\}_{i=1}^{1000} \sim \text{Posterior} \underset{\text{Value Iteration}}{\longrightarrow} \{(\widehat{\pi}^{(i)}, \widehat{Q}^{(i)})\}_{i=1}^{1000}$$
$$\underset{\text{Empirical Mean}}{\longrightarrow} \{(\widehat{\pi}, \widehat{Q})\} \underset{\text{Compute Results}}{\longrightarrow}$$

# 6. Results

**Our approach outperforms the baselines in terms of accuracy on both deterministic-policy and stochastic-policy states.** For deterministic-policy states, the more data the more accurate we are across the settings and the less variability we observe. We also observe that in our method (clipped), the constraints for deterministic-policy states enforce 100% accuracy on such states while the MLE and unclipped method, which does not have such constraints cannot achieve such performance. These results are shown in Table 1. For stochastic-policy states, our introduction of constraints explicitly enforces that we will never pick actions that are $\epsilon$ away from the best action (bad mistakes). In contrast, the MLE and unclipped methods do not have such a guarantee. What is also interesting to observe is that even though our constraints do not say anything explicitly about which action to take in the epsilon balls we see that our method is still more accurate than the MLE and Unclipped when inferring actions for stochastic-policy states. This show how the constraints can help to infer the uncertainty over stochastic-policy states. (See Table 1)

$T^{MLE}$ **makes more bad mistakes, regardless of the amount of data used.** Results from Table 1 show that having more data does not necessarily help in inferring better actions. Specifically, though $T^{MLE}$ uses more data, this does not prevent it from making more bad mistakes (actions not in the $\epsilon$-balls), since certain states will still be hard to infer even with more data. This is due to that the data only covers a small part of the state action space and even in higher data settings, a large part of the state action space will remain unexplored. This is due to aleatoric uncertainty.

**Our constraints produce policies that have considerably less variance across all datasets in comparison to the baselines.** Results from Table 2 and Figure 1 show that in addition to outperforming the MLE and the Unclipped

posterior at inferring the best action, our method (Clipped) has a lot less variance across different datasets thanks to the constraints that enforce the T to recover the $\epsilon$ ball property when planning.

**Our approach enables quantifying the uncertainty and ranking over actions for stochastic-policy states $s$ thus resulting in more informative and performant policies for planning.** The policy and $Q$ values function we get when planning on our Clipped Posterior will quantify the uncertainty in a more informative way by creating a ranking over the actions in the $\epsilon$ ball $\epsilon(s)$ for all the stochastic-policy states s. This is particularly useful when we present such policies to clinicians. The policy from the $T^{MLE}$ will be deterministic even in states where we have uncertainty which can be dangerous as $T^{MLE}$ is prone to making a lot of mistakes, while the policy of the expert has no idea of the ranking of actions in the $\epsilon(s; T)$ in stochastic-policy states s. The uncertainty modeled from the data coupled with our constraints used by the Clipped Posterior, is what helps us infer that ranking while making sure that this uncertainty is correct with respect to the $\epsilon$-ball definition of the expert. It can also help us at inferring a better policy than the one we are given by the expert (See Table 2).

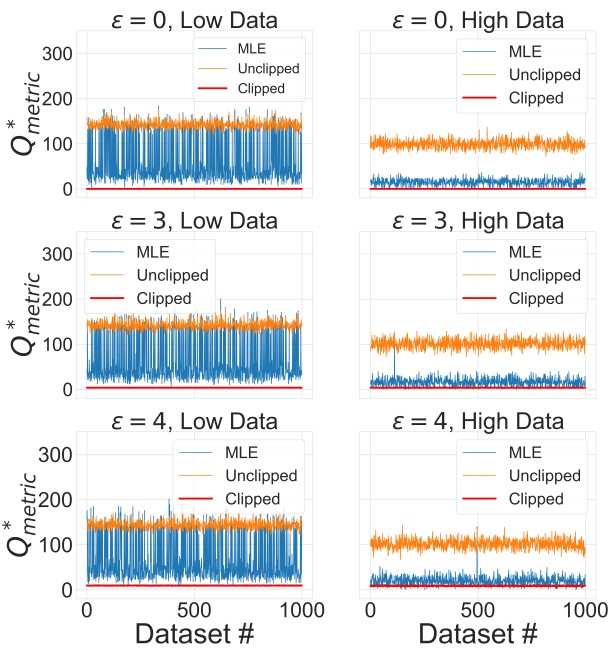

*Figure 1.* Result of the $Q^* metric$ across 1000 different datasets/batch data, where the $Q^* metric$ measure quantitatively how close the inferred policy $\widehat{\pi}$ is to the true unknown policy $\pi^*$. Our method performs the best as well as exhibiting the least amount of variance.

*Table 1.* (Deterministic): accuracy in % of the deterministic-policy states, (Stochastic): the accuracy of the stochastic-policy states ($a \in \epsilon$-balls): the percentage of mistakes that are still $\epsilon$-close to the best action. (When $\epsilon = 0$ there are no stochastic-policy states due to the expert being fully optimal which is why there is no result for the Stochastic and $a \in \epsilon$-balls columns). On each metric, the Clipped posterior achieve better results.

| $\epsilon = 0$: 0 stochastic-policy States | | | |
|---|---|---|---|
| Low Data (Accuracy in %) | | | |
| Method | Deterministic | Stochastic | $a \in \epsilon$-balls |
| Unclipped | 48 ±9 | N/A | N/A |
| MLE | 67 ±1 | N/A | N/A |
| **Clipped** | **100 ± 0** | N/A | N/A |
| High Data (Accuracy in %) | | | |
| Method | Deterministic | Stochastic | $a \in \epsilon$-balls |
| Unclipped | 61 ±6 | N/A | N/A |
| MLE | 83 ±7 | N/A | N/A |
| **Clipped** | **100 ± 0** | N/A | N/A |

| $\epsilon = 3$: 3 Stochastic States | | | |
|---|---|---|---|
| Low Data (Accuracy in %) | | | |
| Method | Deterministic | Stochastic | $a \in \epsilon$-balls |
| Unclipped | 48 ±09 | 37 ±20 | 64 ±37 |
| MLE | 65 ± 10 | 44 ±23 | 83 ±29 |
| **Clipped** | **100 ± 0** | **53 ±25** | **100 ± 0** |
| High Data (Accuracy in %) | | | |
| Method | Deterministic | Stochastic | $a \in \epsilon$-balls |
| Unclipped | 67 ±8 | 33 ±25 | 30 ±30 |
| MLE | 87 ±7 | 53 ±28 | 70 ±38 |
| **Clipped** | **100 ± 0** | **55 ±29** | **100 ± 0** |

| $\epsilon = 4$: 6 Stochastic States | | | |
|---|---|---|---|
| Low Data (Accuracy in %) | | | |
| Method | Deterministic | Stochastic | $a \in \epsilon$-balls |
| Unclipped | 53 ± 9 | 34 ±15 | 81 ±18 |
| MLE | 71 ±11 | 38 ±16 | 87 ±17 |
| **Clipped** | **100 ± 0** | **46 ±18** | **100 ± 0** |
| High Data (Accuracy in %) | | | |
| Method | Deterministic | Stochastic | $a \in \epsilon$-balls |
| Unclipped | 81 ± 7 | 31 ±17 | 38 ±20 |
| MLE | 92 ± 6 | 52 ±20 | 77 ±25 |
| **Clipped** | **100 ± 0** | **59 ±19** | **100 ± 0** |

*Table 2.* Our method (Clipped) can recover a policy even better than the policy of the expert $\pi_b$ which is already near optimal. It recovers such a policy by having very little variance of various batch data. This holds over different degrees of optimality.

| $\epsilon = 0$: 0 stochastic-policy states | |
|---|---|
| Low Data | |
| Method | $Q^*_{metric}$ |
| Unclipped | $142.17 \pm 8.66$ |
| MLE | $59.75 \pm 51.99$ |
| **Clipped** | **0 ±0** |
| Expert | **0 ±0** |
| High Data | |
| Method | $Q^*_{metric}$ |
| Unclipped | $99.0 \pm 9.73$ |
| MLE | $14.75 \pm 6.98$ |
| **Clipped** | **0 ±0** |
| Expert | **0 ±0** |

| $\epsilon = 3$: 3 stochastic-policy States | |
|---|---|
| Low Data | |
| Method | $Q^*_{metric}$ |
| Unclipped | $141.81 \pm 8.92$ |
| MLE | $62.71 \pm 51.32$ |
| **Clipped** | **3.56 ±0.1** |
| Expert | 3.60 ±0 |
| High Data | |
| Method | $Q^*_{metric}$ |
| Unclipped | $100.83 \pm 10.45$ |
| MLE | $16.32 \pm 9$ |
| **Clipped** | **3.56 ± 0.25** |
| Expert | 3.60 ±0 |

| $\epsilon = 4$: 6 stochastic-policy States | |
|---|---|
| Low Data | |
| Method | $Q^*_{metric}$ |
| Unclipped | $143.52 \pm 9.02$ |
| MLE | $65.85 \pm 50.83$ |
| Clipped | $9.45 \pm 0.23$ |
| **Expert** | **9.4 ±0** |
| High Data | |
| Method | $Q^*_{metric}$ |
| Unclipped | $101.54 \pm 10.81$ |
| MLE | $20.11 \pm 9.85$ |
| **Clipped** | **8.79 ± 0.6** |
| Expert | 9.4 ±0 |

# 7. Discussion

We focused on addressing the challenges associated with learning the transition function, $T$, in a gradient-free manner, under offline, tabular, and inverse settings while capturing our desired outcomes. First, using the expert's feedback, we introduced a novel constraint-based approach that explicitly incorporates our desiderata for learning $T$ without relying on gradients. By doing so, we mitigate the limitations and complexities that often arise when gradient-based methods are employed. Second, we investigate the performance of $T$ obtained through MLE, denoted as $T^{MLE}$, across various data and optimality settings. We comprehensively analyze the mistakes made by this $T^{MLE}$ when learning under an uneven coverage dataset due to the expert $\pi_b$ being $\epsilon$-optimal. Third, we demonstrate that by integrating our proposed constraints and incorporating uncertainty, our approach effectively avoids and reduces such mistakes. Consequently, our method outperforms $T^{MLE}$ across different evaluation metrics and in diverse settings. Additionally, we showcase how our constraints significantly decrease the variance of the policy generated from different batch data, even in scenarios where these datasets exhibit substantial variation. Furthermore, we highlights the benefits of combining uncertainty with constraints, particularly in the context of planning. Our results indicate that this combination enables us to derive a ranking of the actions in the $\epsilon$-ball. Consequently, the policies derived from our approach are not only more informative by inferring a policy that is more discriminative in the stochastic-policy states but also demonstrates superior performance in planning tasks. In summary, our contributions include the development of a constraint-based approach for learning $T$ without gradients, a comprehensive analysis of the limitations of $T^{MLE}$, and the successful integration of constraints and uncertainty to enhance the performance and informativeness of policies. Future work could extend this work assuming we are only given the batch data and not the expert policy $\pi_b$ and extend the method to partially observable domains or continuous state and action spaces.

**Acknowledgements** This material is based upon work supported by the National Science Foundation under Grant No. IIS-2007076. Any opinions, findings, and conclusions or recommendations expressed in this material are those of the author(s) and do not necessarily reflect the views of the National Science Foundation.

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

## A. You *can* have an appendix here.

You can have as much text here as you want. The main body must be at most 8 pages long. For the final version, one more page can be added. If you want, you can use an appendix like this one, even using the one-column format.

