# OpenReview forum: "Bayesian Inverse Transition Learning for Offline Settings"
_ICML.cc/2023/Workshop/ILHF — ILHF Workshop ICML 2023_

### Official Review · Reviewer_XgAh · 2023-06-16
**This paper proposes a constraint-based inverse transition learning technique by using expert feedback**

**Rating:** 6
**Confidence:** 3

**Review:**

This paper proposes a new constraint-based approach for inverse transition learning, under expert feedback. The main idea is to first create a set of constraints which ensures that the value function (under estimated T) should be higher for the experts' demonstrations. Then it performs inference on the learned T and estimate the overall transitions. To improve upon that, they use the constraints to clip the posterior through rejection sampling. Experiments is done on synthetic datasets, showcases the constraint-based approach decreases the variance of the learned policy and the benefits of combining uncertainty with constraints in terms of planning.

Overall, this paper presents an interesting idea of exposing explicit constraint for transition learning, by using expert data. The writing could be improved, and more real-world datasets/environments experiment will be helpful for understanding the applicability of the method.

---

### Decision · Program_Chairs · 2023-06-20

Accept